# Latent Action Diffusion for Cross-Embodiment Manipulation

**Abstract:** End-to-end learning approaches offer great potential for robotic manipulation, but their impact is constrained by data scarcity and heterogeneity across different embodiments. In particular, diverse action spaces across different end-effectors create barriers for cross-embodiment learning and skill transfer. We address this challenge through diffusion policies learned in a latent action space that unifies diverse end-effector actions. We first show that we can learn a semantically aligned latent action space for anthropomorphic robotic hands, a human hand, and a parallel jaw gripper via contrastive learning. Second, we show that by using our proposed latent action space for co-training on manipulation data from different end-effectors, we obtain capable policies that can control different robotic embodiments and obtain up to 28% improved manipulation success rates through cross-embodiment skill transfer. Our approach using latent cross-embodiment policies presents a new method to unify different action spaces across embodiments, enabling efficient multi-robot control and data sharing across robot setups. This unified representation significantly reduces the need for extensive data collection for each new robot morphology, accelerates generalization across embodiments, and is an important step towards more scalable and efficient robotic learning.

**Keywords:** Imitation Learning, Cross-Embodiment Learning, Manipulation

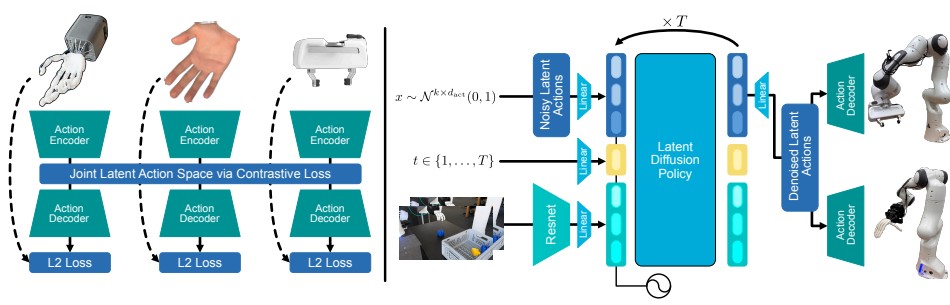

Figure 1: We introduce a framework for learning cross-embodied manipulation policies through latent action diffusion. First, a shared latent action space is discovered through contrastive learning from pairs of aligned end-effector poses. Training diffusion policies with latent actions enables multi-embodiment control with a single policy and realizes skill transfer between embodiments.

## 1 Introduction

Robotic manipulation holds vast transformative potential to address global labor shortages through easy-to-deploy robotic workers able to adapt to different settings. End-to-end learning of manipulation policies is set to equip robots with the necessary skill set and intelligence. The end-to-end learning paradigm has proven its success in high-data regimes for language and vision models.

Submitted to the 9th Conference on Robot Learning (CoRL 2025). Do not distribute.

Robot learning, however, presents novel challenges: imitation learning models are still in a data-bound regime where real-world performance is largely dictated by the volume and diversity of the training data. Scaling up both of these factors inevitably requires pooling together data from different robotic embodiments. However, while it is possible to increase data volume and diversity through cross-embodiment learning, the heterogeneity across observation and action spaces of different robotic embodiments poses significant barriers for skill transfer across embodiments (the "embodiment gap").

Recent works on cross-embodiment learning have largely avoided explicitly addressing the problem of the embodiment gap in action spaces by only using data with a shared action space for pre-/co-training [1, 2, 3]. Other works showing pretraining on human manipulation datasets have relied on explicitly aligning the human action space to the robot action space [4, 5, 6, 7]. In this work, instead of using an explicit action space, we introduce a learned latent action space which can encode diverse action spaces from different end-effectors into a unified, semantically aligned latent action space. To achieve semantic alignment within the latent action space, we utilize retargeting methods, which enable precise alignment of different end-effector action spaces. For policy learning with latent actions, we factorize policies into an embodiment-agnostic policy trained on latent actions and multiple embodiment-specific decoders that are trained separately. Our proposed framework combines the simplicity of training policies with aligned observation and action spaces while still enabling learning from diverse robotic embodiments.

We focus on embodiment transfer among single-arm robots with different end-effectors: the Faive robotic hand [8], the mimic hand [9] and a Franka parallel gripper. Across three experiments, we compare co-trained latent policies against single-embodiment policies. Our findings show that our proposed methodology enables both multi-robot control and positive skill transfer across embodiments with up to 28% (15.2% average) absolute success rate improvement.

Our results indicate the potential of utilizing contrastive learning to bridge heterogeneous action via learned action spaces. As increasingly dexterous, human-like end-effectors become more common, our methodology provides a path forward for effectively sharing and reusing datasets across embodiments with diverse end-effectors through a unified latent action space.

## 2   Methodology

We propose a two-stage framework for cross-embodied latent space imitation learning. First, we learn a shared latent action space via contrastive learning, framing the problem as multimodal representation learning task. Then, we learn policies in the shared latent space via cotraining.

### 2.1   Creating Aligned Action Pairs

Multimodal representation learning architectures for $M$ modalities generally rely on tuples containing paired data of the form $\mathbf{x}_i = \left(x_i^1, x_i^2, \ldots, x_i^M\right)$, where there is some form of cross-modal correspondence between the elements of each tuple [10, 11]. In the context of robotics, our data consists of different end-effector poses that we can align via retargeting functions from human hands to robotic end-effectors:

$$\mathbf{x}_i = (x_i^H, f_H^{R_1}\left(x_i^H\right), \ldots, f_H^{R_M}\left(x_i^H\right)) \tag{1}$$

where $f_H^{R_j}$, $j \in \{1, \ldots, M\}$ are retargeting functions from human hands to the j-th robot embodiment. More details about the retargeting functions are described in Section 6.3.

### 2.2   Contrastive Latent Space Learning

For a shared latent action space, it is crucial that 1) for each modality, sufficient information is encoded such that we can precisely reconstruct end-effector poses and 2) the latent space has a coherent structure, meaning that the cross-modal alignment present in the model inputs during training

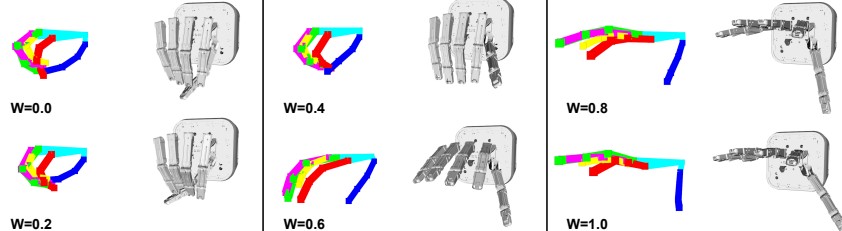

Figure 2: Qualitative evaluation of the joint latent action space. We encode normalized gripper widths $W \in [0, 1]$ (from closed to open) and perform cross-modal reconstruction by decoding them into human hand poses (colored lines on left) and poses for the Faive hand (grey model on right).

is upheld in the learned latent space. To achieve both of these goals, we propose a two-step learning procedure: first, using batches with $B$ aligned end-effector poses, $M$ modality-specific encoders $q_m, m \in 1 \dots M$ are trained that project actions $x_m$ from each end-effector into a shared latent space, where we utilize a pairwise InfoNCE loss [12] to ensure alignment within the batch:

$$\mathcal{L}_{\text{contrastive}} = \frac{1}{M(M-1)} \sum_{i=1}^{M} \sum_{j=i+1}^{M} \left( -\frac{1}{B} \sum_{n=1}^{B} \log \frac{\exp(q_i(x_i^n) \cdot q_j(x_j^n)/\tau)}{\sum_{k=1}^{B} \exp(q_i(x_i^n) \cdot q_j(x_j^k)/\tau)} \right) \quad (2)$$

where $\tau$ denotes the temperature. In the second stage, we train $M$ modality-specific decoders $p_m, m \in 1 \dots M$, which learn to reconstruct ground truth actions $\hat{x}_i$ from their latent representations. Additionally, the encoders $q_m$ are fine-tuned with a lower learning rate. The total loss $\mathcal{L}_{\text{total}}$ backpropagated through the encoders and decoders is a combination of a reconstruction loss $\mathcal{L}_{\text{recon}}$ and the previous contrastive loss $\mathcal{L}_{\text{contrastive}}$, where the hyperparameter $\lambda$ can be used to control the trade-off in between alignment and self-reconstruction:

$$\mathcal{L}_{\text{recon}} = \frac{1}{M} \sum_{i=1}^{M} \sum_{n=1}^{B} ||p_i(q_i(x_i^n)) - \hat{x}_i^n||_2^2 \quad (3)$$

$$\mathcal{L}_{\text{total}} = \mathcal{L}_{\text{recon}} + \lambda \mathcal{L}_{\text{contrastive}} \quad (4)$$

Training details can be found in Section 6.5. We show a qualitative evaluation in Fig. 2.

# 3 Experimental Results

We conduct experiments covering three different end-effectors and three tasks across two setups: one with the Faive hand and the Franka gripper and one setup with the mimic hand and the Franka gripper. In the experiments shown in Fig. 4b and Fig. 3a, we utilize one external camera view as observation, whereas in Fig. 4a, we utilize an additional wrist camera of the mimic hand which is replaced by zero-padding for the Franka hand. For each end-effector in each setup, we compare a single-embodiment diffusion policy with a latent diffusion policy co-trained on data from all end-effectors in each respective setup. Encoders and decoders for the contrastive action model are parameterized by lightweight MLP networks. Additionally, we validate our design choices for our contrastive action model through an ablation study (Table 1).

## 3.1 Discussion and Conclusion

**Latent action representations enable multi-robot control and cross-embodiment skill transfer.** Across all experiments, we find that, a single latent policy can control two highly different end-effectors. Furthermore, latent policies learn helpful shared representations that we observe in

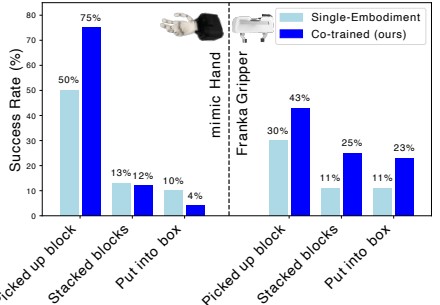

(a) Task stage completion rates for each task stage for single-embodiment diffusion policy versus cross-embodied latent diffusion policy (ours).

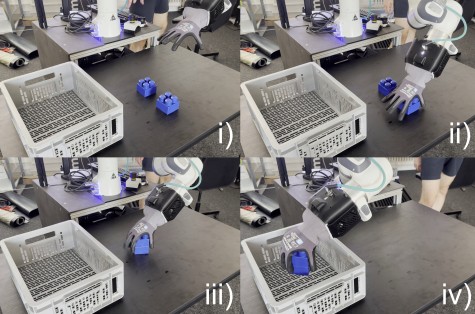

(b) Task stages: i) initial setup ii) picking the first block iii) stacking/inserting it on top of the second block iv) putting both into the box.

Figure 3: We evaluate our methodology across 70 trials per policy per embodiment on a highly challenging block stacking task, comprised of three stages. 200 demonstrations were used per embodiment.

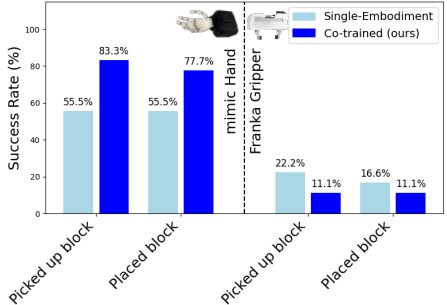

(a) Pick-and-place of a plastic cube into a box in different locations with the mimic hand and Franka gripper using asymmetric observations. 200 demonstrations were used per embodiment.

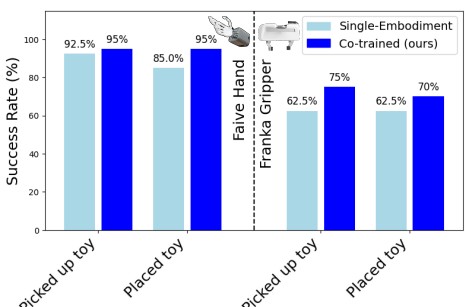

(b) Pick-and-place of a plush toy into a bowl in different locations with the Faive hand and Franka gripper using symmetric observations. 100 demonstrations were used per embodiment.

Figure 4: We compare latent diffusion policies co-trained on cross-embodiment data (ours) with single-embodiment diffusion policies. The task setups are shown in Fig. 5.

improved success rates when co-training on data from different end-effectors for both coarse and fine-grained manipulation tasks (Fig. 3). Further, the benefit of co-training seems to increase with the number of demonstrations utilized, indicating promising scaling behavior.

**Learning with asymmetric observations remains challenging.** We attribute the performance drop of the Franka gripper in Fig. 4a to the asymmetric observations: co-training with missing camera views remains an open challenge (similar to [3]). We hypothesize that the slight performance decrease for the mimic hand in Fig. 3a for fine-grained manipulation could be explained by occlusion of the object by the mimic hand that is not present with the Franka gripper.

## 4   Conclusion

In summary, our approach successfully demonstrates that latent action spaces can unify control across diverse robotic embodiments while enabling improved performance through cross-embodiment skill transfer. The performance improvement of up to 28% (average: 15.2%) indicates that our method facilitates skill transfer between end-effectors with a large embodiment gap and underlines its potential for wider use across a broader range of robot morphologies. Future work includes scaling our method to internet-scale datasets and large-scale cross-embodiment pretraining.

## 5 Limitations

**Grasping Failure Modes**   The most common failure mode is that the policy fails to position the wrist in a suitable position to grasp the object. The positioning of the end-effector is especially important for the comparatively small Franka gripper, which explains the consistently lower manipulation performance compared to the humanoid hands. If the wrist is correctly positioned, grasps are typically successful. For the block stacking task, the most common failure mode lies in failure to align the pins of the block accurately.

**Asymmetric Dataset Sizes**   We find that adding datasets such as BridgeV2 [13] or DexYCB [14] does not yet improve the performance of the policy. While their diverse respective action spaces can be unified through our method, visual differences and the highly asymmetric scale in dataset sizes still present significant challenges for achieving skill transfer via co-training.

**Asymmetric Observations**   Skill transfer in the presence of asymmetric observations (*e.g.* one embodiment has an additional camera view) remains an open challenge for future work, which is reflected in our experiments.

**Ambiguity in Action Space Mapping**   The current contrastive learning method for learning joint action spaces does not automatically guarantee high quality reconstruction for all embodiments. Presently, it is crucial to empirically evaluate the encoders and decoders to verify that reconstruction and cross-reconstruction errors are low, before beginning policy training.

**Latent Space Regularization**   Given that our method for contrastive latent space learning has no implicit latent space regularization (unlike VAE-based methods), there is a risk that the latent space might be suboptimally non-smooth and hard to model for the downstream policy. Future work is needed to guarantee the smoothness of the latent space while maintaining alignment and high reconstruction accuracy.

**Future work: Larger scale experiments and more embodiments**   Present results are still limited to small-scale experiments on a relatively low number of embodiments (albeit a higher diversity of end-effector morphology compared to other cross-embodiment works). We leave larger-scale experiments to future work.

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

## 6 Appendix

### 6.1 Pick and Place Task Setups

We show the rollout environment for the two pick and place tasks in Fig. 5.

### 6.2 Ablation Study: Contrastive Action Model

To validate our design choices, we compare several versions of the contrastive action model (Table 1). As metrics, we utilize self-reconstruction (SR) and cross-reconstruction (CR, Equation (7)) validation losses. The ablation without temperature annealing keeps the temperature constant at the previous final value. The ablation without finetuning freezes the encoders in the second training step while the decoders are being trained. Both temperature annealing and finetuning the encoders substantially improve both self- and cross-reconstruction metrics, with finetuning being the most important addition to the pipeline.

### 6.3 Retargeting

For retargeting, we follow the technique introduced by Sivakumar et al. [15], which utilizes keyvectors for both the human and robot hand. The keyvectors $v_i^{\{H,M\}}(\theta_{\{H,M\}})$ are vectors from the palm to each fingertip and from each fingertip to all other fingertips and provide a unifying representation that can be defined for any hand with a notion of fingertips. For example, to map from human hand poses $\theta_H$ to joint angles for a mimic hand $\theta_M$, we optimize the following differentiable objective using keyvector scaling factors $s_i$:

$$\theta_M(\theta_H) = \text{argmin}_{\theta_F} \sum_{i=1}^{15} \left|\left| v_i^H(\theta_H) - s_i v_i^F(\theta_M) \right|\right|_2^2 \tag{5}$$

For parallel jaw grippers, we take the minimum of all keyvectors originating at the thumb and normalize it by a standard gripper width $W$ such that $\theta_P \in [0, 1]$:

$$\theta_P(\theta_H) = \min_{\theta_P} \left( \min_i \frac{\left|\left| v_i^H(\theta_H) \right|\right|}{W}, 1 \right) \tag{6}$$

### 6.4 Policy Architecture

In the following, we provide more implementation details on our Latent Diffusion Policies.

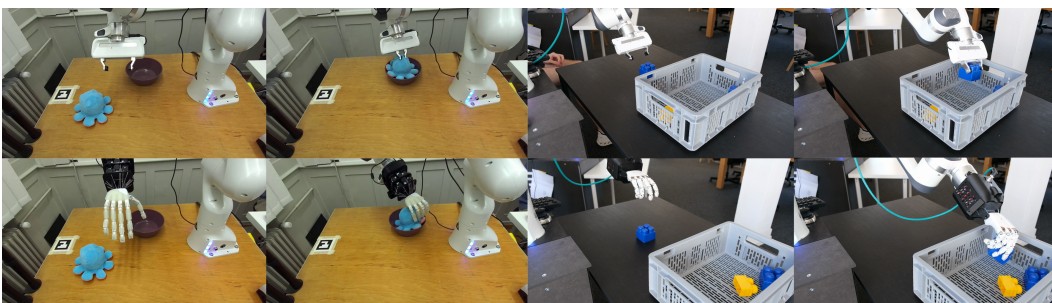

Figure 5: Cross-embodiment policy rollouts for two pick and place tasks in different settings. The robots in each setting (left: Franka gripper, Faive hand, right: Franka gripper, mimic hand) are controlled by a single policy, demonstrating multi-robot control.

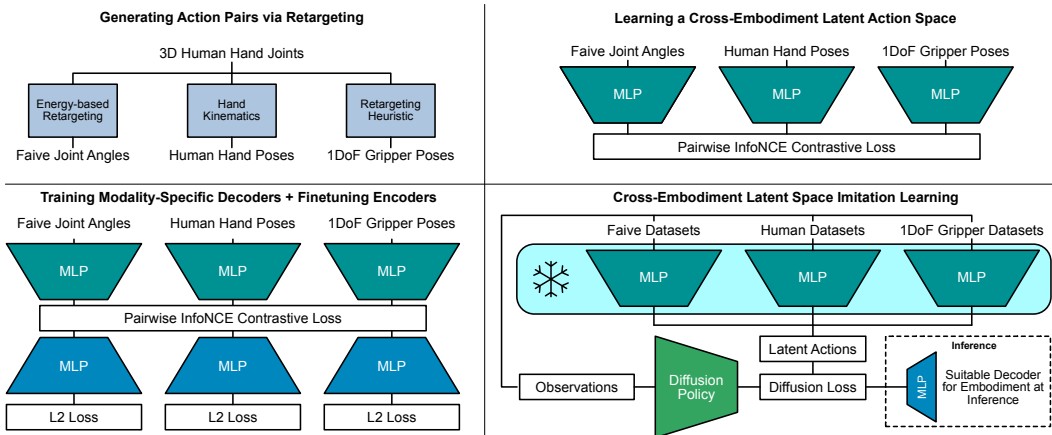

Figure 6: Overview of our proposed framework for semantic alignment of end-effector actions via retargeting (stage 1), training contrastive encoders and decoders for the learned latent action space (stages 2 and 3) and policy learning in latent space (stage 4) with data from the Faive hand, human hands and parallel gripper data. In our experiments, we do not yet utilize human data for policy learning due to challenges with highly asymmetric dataset sizes.

Table 1: Ablation study of the contrastive action model components.

| Model Configuration | SR-Loss | | CR-Loss | |
|---|---|---|---|---|
| | mimic | Franka | mimic→ Franka | Franka→ mimic |
| Full Model (ours) | **0.762** | 3.7e-8 | **0.002** | **214.20** |
| w/o Temperature Annealing (TA) | 0.948 | **1.5e-8** | 0.007 | 286.64 |
| w/o Finetuning (FT) | 44.76 | 2.6e-8 | 0.013 | 391.85 |
| w/o FT and TA | 49.765 | 2.1e-8 | 0.02 | 397.23 |

### 6.4.1 Network Architecture and Observations

**Transformer-based Diffusion Policy** The transformer-based diffusion policy utilized several input tokens that encompass both observations and latent actions. Observations are limited to a single external RGB camera, images of which are encoded by a ResNet18 [16]. The image embeddings, diffusion timestep, and the noisy latent actions are projected into tokens and concatenated with the encoded image representation to assemble the input sequence to the diffusion transformer. Sinusoidal positional embeddings are added to the input sequence. The diffusion objective is applied in the shared latent action space as opposed to the individual explicit action spaces.

**U-Net-based Diffusion Policy** The U-Net-based (U-Net [17]) diffusion policy follows the implementation shown in [18] closely, utilizing FiLM layers to condition the action denoising process on observations. To encode image observations, we use small vision transformer networks pretrained following [19]. Observations for the experimental setting with the mimic hand and the Franka gripper include the arm pose relative to its position at the beginning of each action chunk and an external RGB camera. In Fig. 4a, to investigate learning with asymmetric observations, we utilize an RGB wrist camera for the mimic hand, which is replaced by zero-padding for data collected with the Franka gripper. Such asymmetry in observations often occurs in cross-embodiment settings and provides us with the opportunity to study its impact on skill transfer.

**Contrastive Encoders and Decoders** For the contrastive action model, we utilize standard multi-layer-perceptrons (MLPs) as encoders and decoders. After the input layer, for each hidden layer, we first have a normalization layer, then a linear layer, followed by a ReLU (rectified linear unit) activation and a dropout layer. The hidden layers are followed by a another layer normalization and a linear layer.

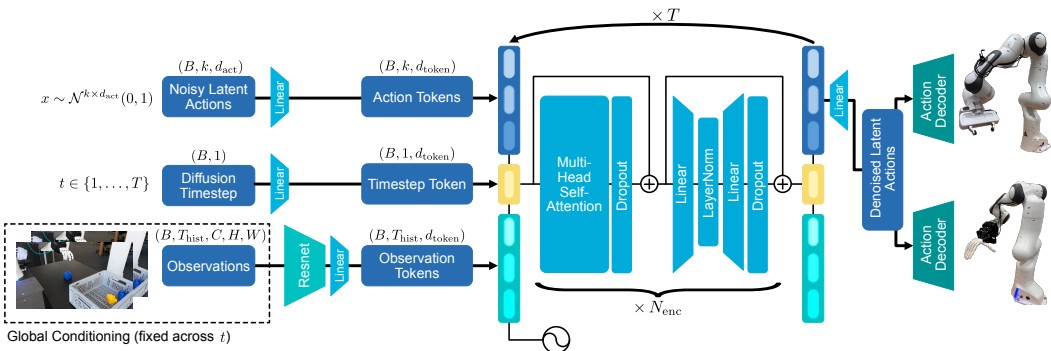

Figure 7: Detailed overview over the policy network architecture for an action chunk size $k$, action dimension $d_{\text{act}}$, an observation history $T_{\text{hist}}$, a token dimension $d_{\text{token}}$, a batch size $B$, and $N_{\text{enc}}$ transformer encoder layers. The observations are fixed as global conditioning during the diffusion process whereas the noisy latent actions are updated with each denoising step.

### 6.4.2  Arm Pose Representation

**Setup #1: Faive Hand and Franka Gripper**   We represent the actions for the arm as deltas $\delta_{\text{arm}} \in \mathbb{R}^6$ in translation and rotation. The delta action for a given timestep $t$ is computed as the difference of the reached poses at time $t + 1$ and $t$. The reached poses are all expressed in the base frame of the arm. For the end-effector poses, we use our proposed latent representation.

**Setup #2: mimic Hand and Franka Gripper**   We follow Chi et al. [20] by representing target arm poses as poses relative to the initial arm pose at the beginning of each action chunk. Target arm poses are obtained from the reached poses of the arm.

### 6.5  Training Details

**Cross-Embodiment Training**   For co-training on differently sized datasets, we assign normalized weights $w_j$ to all datasets. During training, we seek to combine samples from all datasets to fill batches with $B$ samples in total. We sample per-dataset sub-batches with appropriately rounded sizes $\text{round}\left(\frac{B}{w_j}\right)$, project the actions into the shared latent action space, normalize the sub-batches, and then concatenate them into a single batch for efficient training. Through this mechanism, the weight of each dataset approximately represents a sampling probability for each training step. For all experiments, we utilize equal weights for all datasets that are used, such that the model is exposed to the same number of episodes from each embodiment.

**Contrastive Action Model: Human + Faive + Franka**   For training encoders, we found that a batch size of 4096 worked well with a learning rate of 0.001 using the Adam [21] optimizer. We used a weight decay of 0.0001. The temperature followed an exponentially decaying schedule, starting from 0.4 and reducing to 0.2. For training the decoders in a second step, the same hyperparameters were used, but with frozen weights for all encoders. A latent space dimension of 128 worked well to encode 189-dimensional human hand poses, 11-dimensional joint angles for the Faive hand, and 1-dimensional parallel gripper widths. The hidden dimensions for the respective MLP encoders and decoders are 64, 24, and 24. We train the encoders for 300 epochs and the decoders for 50 epochs.

**Contrastive Action Model: mimic + Franka**   For training encoders, we found that a batch size of 16384 worked well with a learning rate of 0.00001 using the AdamW [22] optimizer. We used a weight decay of 0.001. The temperature followed an exponentially decaying schedule, starting from 0.25 and reducing to 0.16. To jointly train the encoders and decoders in the second training stage, the same optimizer and learning rate were used. A latent space dimension of 16 worked well to encode 16-dimensional joint angles for the mimic hand, and 1-dimensional parallel gripper widths.

The hidden dimensions for the MLP encoders and decoders are 32, 128, 128, and 32. We train the encoders for 5000 epochs and the decoders for 10000 epochs.

**Diffusion Transformer: Faive + Franka**   We train our diffusion policies with a batch size of 300 images and their corresponding action chunks with a horizon of 21 timesteps, corresponding to 2.1 seconds. The diffusion noise schedule is a squared cosine schedule with $\beta_{\text{start}} = 0.0001$ and $\beta_{\text{end}} = 0.02$. The learning rate follows a cosine schedule with a warmup with a peak learning rate of 0.0001. We utilize the Adam [21] optimizer over 90k gradient steps for single-embodiment policies and 120k gradient steps for co-trained policies. For co-training on the two similarly sized datasets with the Faive hand and the Franka gripper, we choose equal sampling weights.

**Diffusion U-Net: mimic + Franka**   We train our diffusion policies with a batch size of 256 images for Franka policies with one image observation and a batch size of 128 for cross-embodiment policies with two observations. Action chunks are predicted with a horizon of 48 timesteps, corresponding to 3.2 seconds. The diffusion noise schedule is a squared cosine schedule with $\beta_{\text{start}} = 0.0001$ and $\beta_{\text{end}} = 0.02$. The learning rate follows a cosine schedule with a warmup with a peak learning rate of 0.0001. We utilize the AdamW [22] optimizer, training for 120 epochs for both single-embodiment and co-trained policies. For co-training we also choose equal sampling weights.

## 6.6   Contrastive Action Model: Training Tips

Throughout the development of the model architecture, we came across various qualitative insights for learning latent spaces with these models.

**Latent Space Dimensionality**   We found that in general, it is best to choose the largest size for the latent action space that the downstream policy can fit. This seems to be the most effective way of adding capacity to the action space model, but can conflict with downstream use in policy learning.

**Encoder/Decoder Capacity**   To add encoding and decoding capacity to the models, increasing the depth of the MLPs appears to be more effective than increasing the width. Both help, but we recommend to start increasing depth before width.

**Temperature**   The right temperature choice highly depends on the data that is being fitted. In general, higher temperatures may incur a lower contrastive loss, but can hinder the accuracy of self- and cross-reconstructions. We recommend to sweep over initial and final temperatures to identify values that work well.

**Training Duration**   With larger latent action space dimensions, the models seem to converge faster and require less training. With smaller action spaces, models take significantly longer to train.

## 6.7   Additional Definitions

**Cross-Reconstruction (CR) Loss**   From modality $i$ to $j$, given paired end-effector poses $(x_i^n, x_j^n)$, the CR-Loss is:

$$\mathcal{L}_{\text{CR(i,j)}} = \frac{1}{B} \sum_{n=1}^{B} \left|\left| p_j \left( q_i \left( x_i^n \right) \right) - x_j^n \right|\right|_2^2 \tag{7}$$

We encode data from modality $x_i^n$, decode it to modality $j$ and evaluate the result versus the paired ground truth data $x_j^n$. The self-reconstruction (SR) loss is equivalent to $\mathcal{L}_{\text{CR(i, i)}}$.

