# OpenReview forum: "Latent Action Diffusion for Cross-Embodiment Manipulation"
_robot-learning.org/CoRL/2025/Workshop/Dexterous_Manipulation — CoRL 2025 Workshop Dexterous Manipulation Spotlight_

### Official Review · Reviewer_ZkGP · 2025-09-08
**I recommend to accept this paper**

**Rating:** 7
**Confidence:** 3

**Review:**

This paper presents a method to learn a latent action to deal with hand manipulation with diverse embodiments. To learn a latent action for different embodiments, the authors first propose to learn encoder-decoder structures with a contrastive learning method, followed by the diffusion transformer architecture to learn actions in the latent action space. The paper is well written, easy to follow, and most of the negative comments I can imagine are already mentioned in the limitations. Since this work is well organized, provides many interesting topics to be discussed, and is well-suited for the purpose of the workshop, I recommend accepting this paper.

---

### Official Review · Reviewer_QoHK · 2025-09-11
**This paper introduces Latent Action Diffusion for cross-embodiment robotic manipulation. The authors propose to learn a semantically aligned latent action space via contrastive learning, enabling diffusion policies to operate across diverse end-effectors such as robotic hands, human hands, and grippers. Empirical results show up to 28% improvement in success rates over single-embodiment policies, supported by ablations. Overall, I find the paper novel and relevant, addressing the important challenge of unifying heterogeneous action spaces for scalable robot learning. However, I also have concerns about the limited experimental scope, the unclear role and scale of human data in latent space construction, and the lack of evaluation on more dexterous or asymmetric tasks. Therefore, the paper makes a valuable contribution but requires more validation and clarification to fully establish its impact.**

**Rating:** 8
**Confidence:** 5

**Review:**

Overall, I find the paper novel, well-presented, and addressing an important problem. The idea of unifying heterogeneous action spaces through learned latent representations is promising and could significantly reduce the cost of data collection across robot morphologies. At the same time, I have some questions regarding the role and scale of human hand datasets, the effects of combining different embodiments in training, and the generalization to more dexterous tasks. These considerations suggest further clarification and expansion would strengthen the work.

Pros
1. Addresses an important challenge: the embodiment gap in robotic manipulation.
2. Methodological novelty: latent action alignment via contrastive learning combined with diffusion policies.
3. Empirical improvements: up to 28% higher success rates compared to single-embodiment baselines.
4. Ablation study: validates encoder finetuning and temperature annealing.

Cons
1. Human hand dataset usage: The paper does not specify how large or diverse the human hand dataset needs to be for stable latent space construction.
2. Embodiment combinations: Experiments focus on two-embodiment setups (Faive+Franka, Mimic+Franka). It would be informative to test all three embodiments together, or Faive+Mimic (two anthropomorphic hands), to see whether similarity aids transfer.
3. Task complexity: Only pick-and-place and stacking are tested. It is uncertain how well the method generalizes to dexterous, high-dimensional tasks (e.g., twisting a bottle cap).
4. Scalability: The method has not yet been validated on large-scale datasets with strong asymmetry, limiting claims of generalization.
5. Failure analysis: While some limitations are discussed, a deeper breakdown of when cross-embodiment transfer helps vs. hurts (e.g., embodiment similarity, asymmetric observations) would be valuable.

---

### Decision · Program_Chairs · 2025-09-18

Accept (Spotlight)